# Omega-3 Fatty Acid-Derived Resolvin D2 Regulates Human Placental Vascular Smooth Muscle and Extravillous Trophoblast Activities

**DOI:** 10.3390/ijms20184402

**Published:** 2019-09-07

**Authors:** Arzu Ulu, Prakash K. Sahoo, Ana G. Yuil-Valdes, Maheswari Mukherjee, Matthew Van Ormer, Philma Glora Muthuraj, Maranda Thompson, Ann Anderson Berry, Corrine K. Hanson, Sathish Kumar Natarajan, Tara M. Nordgren

**Affiliations:** 1Division of Biomedical Sciences, School of Medicine, University of California, Riverside, Riverside, CA 92521, USA; 2Department of Nutrition & Health Sciences, University of Nebraska-Lincoln, Lincoln, NE 68583, USA (P.K.S.) (P.G.M.); 3Department of Pathology and Microbiology, University of Nebraska Medical Center, Omaha, NE 68198, USA; 4Cytotechnology Education, College of Allied Health Professions, University of Nebraska Medical Center, Omaha, NE 68198, USA; 5Pediatrics Department, University of Nebraska Medical Center, Omaha, NE 68198, USA (M.V.O.) (M.T.) (A.A.B.); 6Medical Nutrition Education, College of Allied Health Professions, University of Nebraska Medical Center, Omaha, NE 68198, USA

**Keywords:** omega-3 fatty acids, specialized pro-resolving mediators, placenta, Resolvin D2, delivery, GPR18, trophoblasts, inflammation

## Abstract

Omega-3 fatty acids are important to pregnancy and neonatal development and health. One mechanism by which omega-3 fatty acids exert their protective effects is through serving as substrates for the generation of specialized pro-resolving lipid mediators (SPM) that potently limit and resolve inflammatory processes. We recently identified that SPM levels are increased in maternal blood at delivery as compared to umbilical cord blood, suggesting the placenta as a potential site of action for maternal SPM. To explore this hypothesis, we obtained human placental samples and stained for the SPM resolvin D2 (RvD2) receptor GPR18 via immunohistochemistry. In so doing, we identified GPR18 expression in placental vascular smooth muscle and extravillous trophoblasts of the placental tissues. Using in vitro culturing, we confirmed expression of GPR18 in these cell types and further identified that stimulation with RvD2 led to significantly altered responsiveness (cytoskeletal changes and pro-inflammatory cytokine production) to lipopolysaccharide inflammatory stimulation in human umbilical artery smooth muscle cells and placental trophoblasts. Taken together, these findings establish a role for SPM actions in human placental tissue.

## 1. Introduction

Omega-3 fatty acids, including docosahexaenoic acid (DHA), eicosapentaenoic acid (EPA), and docosapentaenoic acid (DPA), play critical roles in pregnancy and infant health. Omega-3 fatty acids are critical for brain and eye development, and adequate intakes have been associated with protection against intrauterine growth restriction, improved neurocognitive outcomes and reduced asthma and allergies in children [1,2]. Despite the importance of these essential fatty acids in pregnancy and fetal health, pregnant women and women of childbearing age consistently fail to meet suggested daily intakes of omega-3 fatty acids [3,4,5,6]. Furthermore, the mechanisms underlying the benefits of these lipids is incompletely understood.

Recent investigations have highlighted the importance of omega-3 fatty acids in modulating inflammation physiology, whereby DHA, EPA, and DPA serve as substrates for the biosynthesis of bioactive lipid mediators, termed specialized pro-resolving mediators (SPM), that actively regulate inflammation resolution and repair processes [7,8,9,10]. Emerging evidence suggests that SPM are critically important in pregnancy and perinatal health; SPM concentrations in breastmilk during the first month of lactation are reported to be log-folds higher than circulating SPM levels in healthy adult blood [11,12,13]. Recent studies have also identified SPM intermediates in maternal blood and placental tissues with maternal supplementation leading to increased concentrations of both the omega-3 fatty acid as well as certain intermediate levels [14,15]. Importantly, one investigation identified that SPM levels were lower in mothers with mastitis compared to healthy mothers, which also corresponded with elevated inflammatory lipid metabolites in milk from the mothers with mastitis [12]. These breastmilk-derived SPM were also found to be highly bioactive with potent anti-inflammatory and pro-resolving actions when used as treatments in vitro and in vivo [12]. Furthermore, we recently found increased levels of SPM in maternal and cord blood at the time of delivery, with high RvD1 and RvD2 levels associated with adverse clinical outcomes [16]. Of interest, RvD1 and RvD2 levels were significantly elevated in maternal blood as compared to cord blood, suggesting the placenta as a potential site of action for maternal circulating SPM [16].

SPM exert their potent anti-inflammatory and pro-resolution actions through binding G-protein coupled receptors and initiating or altering intracellular signaling [8]. To investigate the potential role of the placenta as a target for SPM signaling, we performed immunohistochemistry for the RvD2 receptor, GPR18, in human placental tissues. Herein, we identified striking expression of GPR18 in human placental tissue, with expression found in distinct cell types, including placental extravillous trophoblasts (EVT) and decidual cells as well as placental vascular smooth muscle cells. Furthermore, using in vitro cell cultures, we identified that these cells are responsive to SPM, with RvD2 treatment leading to significant alterations in the physiological responses of these cells to exogenous stimuli. Taken together, our results indicate that the placenta is a site of action for SPM, with future studies warranted to better understand the mechanisms and associations between omega-3 fatty acid supplementation, SPM production, and maternal-fetal health outcomes.

## 2. Results

### 2.1. Participant Characteristics

Human placental samples were collected following delivery from mothers in their third trimester at time of delivery. Exclusion criteria included infants with congenital abnormalities, inborn errors of metabolism, gastrointestinal, liver, or kidney disease, and neonates that were anemic or needed blood transfusions. Infants whose parents were less than 19 years of age were also excluded from the study to avoid obtaining parental consent from a parent who is not legal age in the state of Nebraska. Infants who were made wards of the State of Nebraska were also excluded from the study, per Nebraska State law 390 NAC 11-002.04K which states wards of the state may not participate in medical research unless a specific exception is granted by the state, following an evaluation of the protocol by staff in the Medical and Legal Divisions of the Nebraska Health and Human Services System. For placenta collection, damaged or disrupted placentas were excluded. One birth had suspected or confirmed chorioamnionitis, was born at 32 weeks gestational age, and was admitted to the neonatal intensive care unit (NICU). Infants from five additional participants were also admitted to the NICU upon birth. Thirty-five percent of mothers reported taking an omega-3 fatty acid-containing prenatal supplement. Participant characteristics are provided in Table 1.

### 2.2. GPR18 Expression and Localization in Human Placental Tissue

All 26 human placental cross-sections assessed exhibited GPR18 expression. GPR18 expression was most consistently expressed in EVT and decidual cells (Figure 1), and placental vascular smooth muscle cells (Figure 2), with staining also found in trophoblasts and Hofbauer cells of the placental tissue. Given these initial findings, we performed additional analyses to assess protein expression intensity in the placental EVT and vascular smooth muscle cells. After digital image annotation, three placental samples for each analysis were excluded based on criteria described in the methodology (samples 3, 19, and 25 for EVT; samples 2, 15, 24 for vascular smooth muscle). Using Definiens Tissue Studio^®®^ software to assess relative staining intensity in the remaining samples, we found the majority of cells exhibited medium or high GPR18 membranous and cytoplasmic expression for EVT (Figure 1A–C). Twenty-six percent of the samples had no areas of low expression. Thirteen out of twenty-three samples had areas of medium expression that were greater than 50%. Areas of low expression ranged from 0–87%, while areas of medium expression ranged from 12–97%, and high expression ranged from 0–75% (Figure 1D). No significant difference of expression was observed based on maternal and gestational ages.

Vascular smooth muscle cell GPR18 immunoreactivity was also measured in 23 placental tissue samples (Figure 2). As indicated in the methods, three cases were excluded from the analyses (2 had less than 10 blood vessels and one exhibited poor staining). Areas of low expression ranged from 0–91%, while areas of medium expression ranged from 8–100%, and high expression ranged from 0–41% (Figure 2D). Eight cases exhibited more than 50% of low expression. 15 cases exhibited more than 50% of medium to high expression. No significant difference of GPR18 expression in vascular smooth muscle cells was observed when considering maternal and gestational ages.

### 2.3. Confirmation of GPR18 Expression at Transcript Level in Placental Cells

This is to our knowledge the first identification of GPR18 protein expression in placental tissue. To verify expression was not artifactual, we assessed the transcript-level gene expression of *GPR18* in placental cell lines, including normal human immortalized placental trophoblasts (HTR-8), and in choriocarcinoma-derived placental trophoblasts (JEG-3 and JAR cells). As shown in Figure 3, we identified variability in expression of *GPR18* across the cell lines with highest expression in the JAR cell line, which exhibited approximately 5 to 6-fold increased transcript level expression over NTERA (human testicular-derived cell line) and Huh-7 (human hepatocellular carcinoma) and ~2-fold increase over the HepG2 (human hepatocellular carcinomas) cell line. These findings provide transcript-level confirmation of *GPR18* in placental cells that support our protein-level expression findings. Further, we also observed increased *GPR18* expression in choriocarcinoma-derived cells compared to human immortalized cells (Figure 3).

### 2.4. Effects of RvD2 on Placental Trophoblast GPR18 Expression and Function

As the known ligand for GPR18, we assessed whether human placental trophoblasts were responsive to RvD2 treatment. As shown in Figure 4, all placental trophoblasts cells such as HTR-8, JAR, JEG-3 and BeWo exhibited protein-level expression of GPR18 (Figure 4A) and this expression was unchanged when trophoblasts were treated with 100 nM RvD2 for 24 h. Similarly, we identified no significant difference in transcript level expression of GPR18 at 24 h following 100 nM RvD2 treatment (Figure 4B). Further, we also observed significantly increased GPR18 mRNA expression in choriocarcinoma-derived placental trophoblast (JEG-3) cells compared to human immortalized placental trophoblasts, HTR-8 (Figure 4A).

To determine RvD2 receptor binding, we next assessed for GPR18 membrane movement to plasma membrane with RvD2 treatment. As depicted in Figure 5, JAR cells exhibited perinuclear expression of GPR18 in control and RvD2-treated cells. Additionally, we also observed increased membrane localization of GPR18 in JAR cells with RvD2 treatment compared to control (untreated) cells (Figure 5). Furthermore, when HTR-8 cells were treated with the inflammatory stimulus TNFα (60 ng/mL) for 24 h, HTR-8 cells exhibited increased levels IL-1β and IL-6 mRNA with TNFα treatment compared to controls (Figure 6A,B). However, there was a non-significant trend in decreased levels of IL-1β and IL-6 with RvD2 and TNFα co-treated placental trophoblasts. Treatment of RvD2 to unstimulated (no TNFα treatment) cells did not alter the levels of IL-1β and IL-6 mRNA expression (Figure 6A,B). Of interest, in cells not stimulated with TNFα, RvD2 treatment elicited enhanced production of the anti-inflammatory cytokine IL-10 mRNA expression, but this upregulation was lost in TNFα-stimulated cells (Figure 6C). Further, we also found a significant decrease in the levels of IL-18 mRNA expression with TNFα treatment and this decrease in IL-18 expression was unaltered with RvD2 co-treatment (Figure 6D).

### 2.5. Effects of RvD2 on Human Umbilical Artery Smooth Muscle Cell Inflammatory Response

To verify the protein-level expression of GPR18 in placental vascular smooth muscle in vitro, we first performed immunofluorescence staining of primary human umbilical artery smooth muscle cells (HUASMC) grown in culture. As shown in Figure 7, we identified GPR18 protein-level expression in HUASMC. Furthermore, GPR18 protein expression in HUASMC significantly increased following 1-h pretreatment with RvD2 (0–100 nM; Figure 7A,B) or with DHA (0 or 1 µM; Figure 7C) ± 24 h stimulation with inflammatory mediators LPS (0 or 100 ng/mL; Figure 7A) or a cytomix cocktail containing IL-1β (1 µg/mL), TNFα (10 ng/mL) and IFN-γ (1000 U/mL) (Figure 7B,C). Consequently, 1 and 100 nM RvD2 pretreatments also resulted in significant reductions in LPS-induced pro-inflammatory IL-6 release at this time point (Figure 8). Furthermore, HUASMC pretreated with RvD2 (100 nM) exhibited morphological alterations, with significantly increased α-smooth muscle cell actin (αSMA) expression that remained elevated following stimulation with 100 ng/mL LPS (Figure 9A,B). Interestingly, despite these RvD2-mediated changes in cytoskeletal dynamics, we observed no alterations in HUASMC contractility in response to RvD2 in the presence or absence of LPS, as measured via collagen gel contraction assays (Figure 9C,D).

## 3. Discussion

An inflammatory intrauterine environment is a known risk factor for negative pregnancy and birth outcomes, including complications of preterm birth such as chronic lung disease and retinopathy of prematurity [17,18]. Interestingly, the importance of omega-3 fatty acids during pregnancy for both maternal and fetal health is recognized [1,2], but remains incompletely understood. The identification that omega-3 fatty acids are metabolized into bioactive pro-inflammation resolution lipids, SPM [7], provides a potential link between the known benefits of omega-3 fatty acid supplementation in pregnancy and their conferred protection from negative inflammation-associated perinatal outcomes. Of note, we recently identified that maternal circulating blood collected at delivery has significantly higher levels of SPM RvD1 and RvD2 compared to cord blood, suggesting that the placenta could be a site of action for maternal SPM [16]. Through the investigations described herein, we have identified the placenta as a site of action for SPM, with key actions on specific placental cell types including placental vascular smooth muscle and EVT that could confer protection against inflammatory injury and disease.

Recent investigations have also identified elevated biosynthetic intermediates of SPM in maternal blood and placental tissues in mothers supplementing with omega-3 fatty acids [14,15]. For example, in a secondary analyses of the mothers, Omega-3 and Mental Health study, investigators identified that 17-HDHA (17-hydroxy-docosahexaenoic acid; resolvin D series intermediate) levels were increased in maternal blood at 34–36 weeks gestation as compared to collected blood at enrollment (12–20 weeks gestation), and mothers who received EPA or DHA-containing fish oil supplementation exhibited increased 17-HDHA compared to placebo-supplemented mothers [14]. Furthermore, in a study assessing how omega-3 fatty acid supplementation alters placental tissue omega-3 fatty acid and SPM composition, it was found that fish oil supplementation led to increased DHA, 18-HEPE (18-hydroxy-eicosapentaenoic acid; EPA-derived resolvin series intermediate), and 17-HDHA placental levels, but did not significantly alter SPM RvD1, RvD2, or protectin D1 [15]. Also, a recent study identified that supplementation of omega-3 fatty acids during pregnancy increases precursors of SPM in newborns [19]. Adding to these studies are our recent findings that RvD1 and RvD2 levels were significantly elevated in maternal blood collected at labor as compared to cord blood, suggesting the placenta as a potential site of action for maternal circulating SPM [16].

In support of this hypothesis, in our investigations described herein we identified that two cell types with the most consistent GPR18 were the EVT and placental vascular smooth muscle cells. Of note, Hofbauer cells also exhibited GPR18 as is expected given the previously documented expression of GPR18 on macrophages [9]. With regards to different cell types, a study identified an association between a decrease in cervical infiltrating macrophages and decrease in pro-inflammatory cytokine expression in LPS-challenged fat-1 mice, capable of converting omega-6 to omega-3 fatty acids. This study identified EPA-derived Resolvin E3 to be a potential therapeutic in preterm birth [20]. In the context of the role of GPR18 as a receptor for the SPM RvD2, the expression of GPR18 on the vascular-associated cell types is of physiological significance, because appropriate vascular functioning is critical to maintain maternal-fetal health and prevent negative perinatal outcomes. Dysfunctions in placental vasculature increase risk for intrauterine growth restriction, preterm delivery, and pre-eclampsia [21,22,23,24]. While some studies aimed at finding molecular biomarkers for these complications, others focus on lipids. A recent study determined the levels of endothelial progenitor cell and NK cell markers during the first trimester of pregnant women and followed during pregnancy. Markers associated with these cell markers were identified useful during progression to preeclampsia in these women [25]. Other studies focused on epigenetic changes and found a consistency in unique miRNAs presence in placental tissue as well as in maternal serum [26,27]. With regards to anti-inflammatory and analgesic effects of RvD2 [28], one study showed that RvD2 reduces pain via regulating pro-inflammatory cytokines and Akt/GSK signaling pathway. In addition, GPR18 is expressed in neurons and astrocytes suggesting the sites of action for RvD2 [29].

A recent study investigated over 20,000 births, including over 4000 preterm births, to identify placental pathologies associated with adverse neonatal outcomes [23]. In this investigation, placental malperfusion was the most commonly identified pathology found in preterm placentas (including spontaneous preterm birth), at over 50%, while regions of inflammation/infection were also identified in over 25% of preterm placentas. In addition, elevated Resolvin D1 levels are associated with term delivery as compared to preterm delivery in women and in pregnant mice, which involved DHA-mediated inhibition of inflammasome activation in trophoblasts [30]. There is emerging evidence that bioactive lipids can be used as biomarkers in preeclampsia. A study analyzed serum lipid biomarkers from women at 12–14 weeks pregnancy and followed for the duration of their pregnancy. This study was able to identify a panel of lipid biomarkers that can distinguish between controls and preeclampsia cases [31].

Fetal-derived EVT are important early modulators of placental vascular development. EVT invade maternal decidual spiral arteries to increase blood flow from mother to fetus, and defective functioning of these cells contribute to numerous negative perinatal outcomes, including preterm birth [32,33]. Inflammatory stimuli have been shown to reduce invasive capacities of EVT, while inducing their release of pro-inflammatory cytokines [34]. These findings suggest a mechanism by which an inflammatory placental environment may lead to placental vascular dysfunction leading to negative pregnancy outcomes. In this capacity, expression of GPR18 on EVT to confer RvD2-responsiveness could be providing a protective benefit for these cells, making them less responsive or affected by inflammatory stimuli, and this hypothesis warrants further investigation.

Furthermore, the placental vasculature is thought to exist at near maximal vasodilation under healthy conditions [35]. Changes in the function of placental vascular smooth muscle, including changes in differentiation, viability, and contractility, have been identified as predictors for reduced circulation, ischemia, and negative pregnancy and neonatal outcomes [24,36,37]. Interestingly, in our investigations we found that HUASMC have sustained expression of GPR18 under varied inflammatory stimuli (LPS or cytomix cocktail), and that treatment with RvD2 led to reduced responsiveness to LPS stimulation as measured by IL-6 inflammatory cytokine release. In addition, RvD2 treatment led to significant alteration in cytoskeletal dynamics with increased αSMA expression that was sustained during LPS stimulation. Yet, RvD2 treatment had no significant effect on collagen contraction, suggesting these changes in cytoskeletal dynamics are not leading to vascular constriction that could negatively impact fetal blood flow.

In the present study, the expression of GPR18, a receptor for RvD2 was observed in placental trophoblasts, suggesting that omega-3 fatty acid metabolites have a functional role in the placenta. The addition of RvD2 to placental trophoblasts led to increased IL-10 expression indicating that RvD2 acts as a cellular anti-inflammatory nutrient compound similar to its effect observed in hypothalamus against acute inflammation during obesity [38]. Although we did not observe changes in the expression of GPR18 with RvD2 treatment in placental trophoblasts, we saw increased membrane localization in cells with RvD2 treatment and observed a trend towards inhibition of pro-inflammatory cytokine production in TNFα-stimulated placental trophoblasts. Further studies are underway to confirm the role of pertussis-sensitive G-protein coupled receptor, GPR-18 activation [9] in placental trophoblasts with RvD2 treatment during inflammatory stimuli. While our findings provide insight on the role of SPMs in placental tissue, our study is not without limitations. This is intended as a cross sectional study mainly focusing on the assessment of RvD2 and its receptor in trophoblasts and vascular smooth muscle cells of the placenta; however, the fluctuations in SPMs during different trimesters of pregnancy and how this might contribute to fetal health is unknown. Also, while we show in placental cells that RvD2 is involved in reducing inflammation, we are unable to explain any causal relationships regarding RvD2 and preterm births. Further studies are underway in elucidating the protective role for SPMs in the placenta of preterm, preeclampsia and maternal obesity.

Interestingly, omega-3 fatty acids and SPM do have known benefits in cardiovascular functioning including specific actions on vascular smooth muscle cells to reduce responsiveness to inflammatory stimuli [39,40,41,42,43]. These findings corroborate our identification of placental vascular smooth muscle cells as targets for SPM, while our findings detailed in this report add to this knowledge, identifying that HUASMC are responsive to DHA-derived RvD2, including in the setting of pro-inflammatory stimulation.

## 4. Materials and Methods

### 4.1. Materials

Resolvin D2 (7(S), 16(R), 17(S)-Resolvin D2, # 10007279) was purchased from Cayman Chemical (Ann Arbor, MI, USA). LPS was purchased from Sigma-Aldrich (St. Louis, MO, USA). TRIzol reagent and random hexamers were obtained from Invitrogen (Waltham, MA, USA). Protease inhibitor cocktail and SYBR Green master mix was purchased from Roche, Indianapolis, IN, USA. Anti-GPR 18 antibody were obtained from Invitrogen, (# PA5-23218) and HRP-conjugated secondary antibody were from Jackson Immuno Research, West Grove, PA, USA.

### 4.2. Human Placental Tissue Collection

All human placental samples used in these investigations were collected under IRB approval from the University of Nebraska Medical Center (IRB #112-15-EP, Fatty Acids, Fat Soluble Vitamins, Infant Feeding, and Inflammation during NICU Hospitalization. Original approval date 1 April 2016, UNMC IRB Ethics Committee). Written informed consent was received from each participant.

Third-trimester placental tissues were collected from a transitionary post-partum storage fridge within 12 h of delivery, dissected by study personnel into representative cross-sectional samples, and fixed in 10% formalin. Placental tissues in need of a clinical pathology exam had their sample taken from the periphery of the placental disk to avoid interfering with clinically indicated assessments. In total, 26 placental tissues were used in these studies. The demographics and history of the mothers are given in the Table 1.

### 4.3. Placental Tissue Processing and Staining for GPR18

Placental cross-sectional tissues were fixed with formalin and embedded in paraffin wax. Once the four-micron cross sections of 26 tissues was obtained and the glass slides were prepared, the samples were stained with hematoxylin and eosin. Immunohistochemical staining for GPR18 (polyclonal; Thermo Fischer Scientific; 1:75 dilution) was performed on the glass slides using standard staining protocols using a Ventana Discovery Ultra autostainer (Roche, Indianapolis, IN, USA). The glass slides were then digitized at a single focal plane level at 40× magnification using a VENTANA iScan HT slide scanner (Roche, Indianapolis, IN, USA).

Two copies of the 26 digitized images were made and saved in 2 different folders in a password protected encrypted external hard drive for the purpose of analyzing smooth muscle cells and EVT. One set of 26 images were screened and smooth muscles were annotated using the VENTANA Image Viewer software (version 3.1.3; Roche, Indianapolis, IN, USA). Three cases were excluded from the analyses due to having less than 10 blood vessels (2 samples) or poor staining (1 sample). Using Definiens Tissue Studio^®®^ software, the remaining digitized tissue slides with at least ten blood vessels (*n* = 23) were selected to further quantify the percentage (0–100%) and intensity (low vs. medium vs. high) of GPR18 immunoreactivity in smooth muscle cells. Similarly, the second set of 26 images were screened and EVT were annotated. Three EVT samples were excluded because total EVT were fewer than 100 cells across the slide. For EVT and decidual cells, the digitized tissue slides with at least 100 cells were selected to further quantify the percentage (0–100%) and intensity (low vs. medium vs. high) of GPR18 immunoreactivity.

### 4.4. Tissue Culture and Cell Treatments

Human umbilical artery smooth muscle cells (HUASMC, PromoCell GmBH, Heidelberg, Germany; #12500) were cultured in 1:1 Dulbecco’s modified Eagle’s medium (DMEM) with high glucose and glutamine (HyClone via Thermo Fisher Scientific, Waltham, MA, USA) and DMEM F12, supplemented with 10% FBS and Penicillin/treptomycin (10,000 units/mL, Thermo Fisher Scientific, Waltham, MA, USA). Cells were grown to 85% confluency at 37 °C/5% CO_2_.

NTERA an epithelial-like testicular carcinoma cell line was used as a control for GPR18 expression analysis. HTR-8SV/neo (HTR-8) normal human immortalized first-trimester placental trophoblast cells, choriocarcinoma-derived third-trimester placental trophoblast cell lines (BeWo, JAR and JEG-3) and hepatoma-derived hepatocytes (Huh7 and HepG2 cells) were used. Hepatocytes, BeWo, JAR and HTR-8 cells were cultured in DMEM supplemented with 10% fetal bovine serum (FBS), 0.1% Geneticin (G418) and 0.01% Plasmocin. JEG-3 and NTERA were cultured in MEM supplemented with 10% FBS and 0.01% Plasmocin. Cells were treated with 100 nM of RvD2 for 24 h to observe changes in expression level of GPR18. Immunomodulatory effects of RvD2 were studied by inducing the cells with 60 ng/mL of human tumor necrosis factor-α (TNF-α, ab9642, Abcam, Cambridge, United Kingdom) to measure the levels of pro-inflammatory cytokine levels by real-time quantitative PCR.

### 4.5. Human IL-6 ELISA

HUASMC cells were pretreated for one hour with Resolvin-D2 (0.1, 1, 10 and 100 nM) at indicated concentrations followed by 100 ng/mL LPS treatment for 24 h. The next day, supernates were collected for human IL-6 ELISA measurements (DY206, R & D Systems, Minneapolis, MN, USA). Two biological replicates were run for each of three experiments.

### 4.6. Western Blotting

Monolayer of cells seeded in 6-well plates were treated with RvD2 (100 nM) and vehicle for 24 h, then lysed using lysis buffer containing 50 mM Tris pH 7.4, 150 mM NaCl, 1 mM EDTA, 1 mM DTT, 1 mM Na3VO4, 1 mM PMSF, 100 mM NaF, 1% Triton x-100) with the addition complete protease inhibitor cocktail (Roche, Indianapolis, IN, USA). Protein concentration was then determined by Pierce 660 nm Reagent (Thermo Fisher Scientific, Waltham, MA, USA; Catalog # 22660) with BSA as standard. Placental cell lysates containing 30 ug of protein were resolved in SDS-PAGE and the proteins were transferred onto nitrocellulose membrane (BioRad Laboratories, Hercules, CA, USA) for immunodetection using anti-GPR 18 antibody as described [44].

### 4.7. Real-Time PCR

Total RNA was extracted from cells grown in monolayer using TRIzol reagent as per manufacturer’s protocol. One µg of total RNA was used to synthesize cDNA using Random Hexamers and M-MuLV reverse transcriptase (New England Biolabs, Ipswich, MA, USA; Catalog # M0253S) in a 20 µL reaction system. A CFX Connect Real-Time PCR Detection System (BioRad Laboratories, Hercules, CA, USA) was used to perform quantitative PCR in a 20 µL reaction system, including 2 µL primers (10 µM), 1 µL cDNA, 10 µL SYBR Green I Master Mix (Roche, Indianapolis, IN, USA) and molecular biology grade water. Relative expression of mRNAs was normalized against 18S rRNA allowing comparison of mRNA levels and real-time PCR purity was monitored by melting curve analysis. The following primers were used: GPR18, forward: 5′-CCA CCA AGA AGA GAA CCA C-3′; reverse: 5′-GAA GGG CAT AAA GCA GAC G-3′; IL-1b, forward: 5′-CTC GCC AGT GAA ATG ATG GCT-3′; reverse: 5′-GTC GGA GAT TCG TAG CTG GAT- 3′, IL-6, forward: 5′-CTT CYC CAC AAG CGC CTTC-3′; reverse: 5′-CAG GCA ACA CCA GGA GCA-3′; IL-10, forward: 5′-AAGCCTGACCACGCTTTCTA-3′; reverse: 5′-GCTCCCTGGTTTCTCTTCCT-3′, IL-18, forward: 5′-GCT CTG TGT GAA GGT GCA GTT-3′; reverse: 5′-AAT TTC TGT GTT GGC GCA GT-3‘; 18S, forward: 5′-CGT TCT TAG TTG GTG GAG CG-3′; reverse: 5′-CGC TGA GCC AGT CAG TGT AG-3′. Relative RNA expression levels were determined by the delta Ct (2^−ΔCt^) method.

### 4.8. Immunofluorescent Staining

HUASMC cells were grown on glass coverslips overnight. The next day cells were pretreated for one hour with Resolvin D2 followed by 24-h treatment with LPS at 100 ng/mL. At the end of the treatments, cells were fixed in 10% formalin at room temperature and then permeabilized in 0.2% Triton X-100 in PBS for 5 min. Afterwards, the cells were rinsed with PBST (0.1% Tween-20 in PBS), and incubated overnight at 4 °C with antibodies against rabbit alpha smooth muscle cell actin (Abcam, Cambridge, United Kingdom; ab5694) or rabbit anti-GPCR GPR18 antibody (Abcam, Cambridge, United Kingdom; ab76258). The next day, cells were washed with PBST for 5 min three times and incubated with secondary antibodies; goat anti-rabbit Alexa Fluor 488 secondary antibody for 2 h at room temperature. After washing in PBST for 5 min three times, cells were mounted using Prolong Gold antifade reagent with DAPI. Cells were visualized with Echo Revolve epifluorescence microscope (San Diego, CA, USA) at 20× magnification, and data were collected from at least five fields per glass coverslip. Image analysis was performed by measuring fluorescent intensity divided by the area of each cell using Fiji open source software, as described before [45,46,47], and values from each cell in each field were averaged to obtain at least five data points per treatment in each of the three separate experiments.

### 4.9. Collagen Gel Contraction Assay

Rat tail type I collagen (Gibco via Thermo Fisher Scientific, Waltham, MA, USA; A10483-01) was used to prepare the collagen gel as per manufacturer’s instructions. The final concentration of the collagen gel was 1.25 mg/mL. The pH of the collagen gel was monitored using pH strips and it was adjusted to 7.2 with 0.5 M NaOH. Collagen gel contraction assay was performed as previously described (Benoit et al., [48]). Briefly, HUASMCs were harvested by trypsinization, centrifuged at 1500 rpm for 5 min and resuspended in complete media (DMEM:F12) for cell counting. To determine the optimal conditions (duration and cell number), a set of preliminary experiments were performed at 5 × 10^4^, 10 × 10^4^ and 30 × 10^4^ cells/mL and at 12 and 24 h. We determined the optimal conditions to be 30 × 10^4^ cells/mL for 24 h. For all the experiments, cell-collagen gel mixture was cultured for 24 h at 30 × 10^4^ cells/mL concentration. A 500 µL HUASMC/collagen mixture at 30 × 10^4^ cells/mL was plated into a 24-well plate. The plates were polymerized in the tissue culture incubator at 37 °C for 30 min, then 1 mL of complete media (DMEM:F12) was added to each well together with the appropriate treatments (1 h pretreatment with RvD2 followed by 100 ng/mL LPS for 24 h). The gels were lifted using a L-shaped metal spatula to allow the polymerized cell/gel mixture to float freely, and the plates were cultured for 24 h. At the end of each experiment, plates were imaged using Keyence inverted fluorescence microscope (BZ-X710) at 2× magnification and 9 images were obtained per well. Images were then stitched using the Keyence Analyzer ‘Image Stitching Module’ and the area of each stitched image was measured in ImageJ.

### 4.10. Statistical Analysis

All statistical analysis tests were calculated using GraphPad Prism 7 software (GraphPad Software, San Diego, California, USA). A Shapiro-Wilk normality test was performed, and normally distributed datasets were analyzed using unpaired *t*–test/one-way analysis of variance (ANOVA). Those datasets that did not pass the normality test were analyzed using Wilcoxon-Mann-Whitney or Kruskal-Wallis non-parametric test. Post-hoc multiple comparisons were performed using Tukey and Bonferroni methods.

## 5. Conclusions

Taken together, our results identify a role for SPM in altering placental function, with numerous cell types in the placenta expressing the SPM RvD2 receptor GPR18. These findings warrant further studies to ascertain the role for SPM in modulating placental vascular responses during development and under inflammatory conditions. Furthermore, these studies highlight a potential mechanism by which omega-3 fatty acid supplementation during pregnancy may yield protective effects to the developing fetus through modulating placental trophoblast and vascular function.

## Figures and Tables

**Figure 1 ijms-20-04402-f001:**
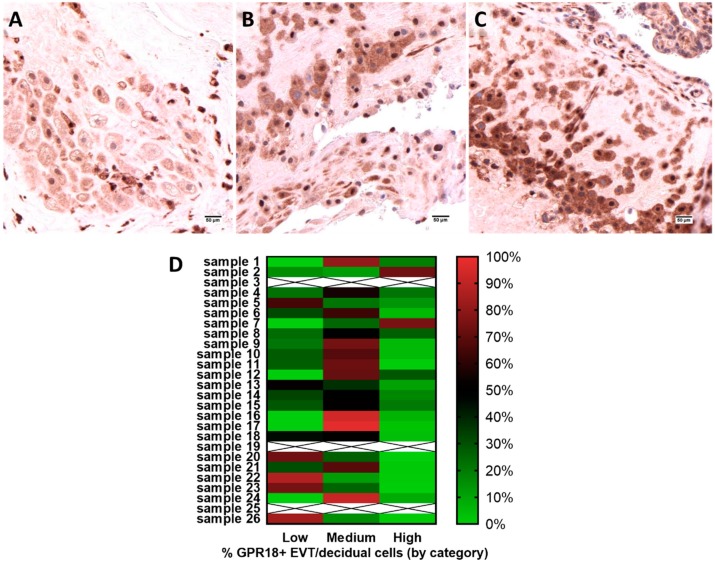
GPR-18 expression in placental extravillous trophoblasts (EVT). Low (**A**), medium (**B**) and high (**C**) expression of GPR-18 in vascular smooth muscle cells of human term placenta. The images shown are representative, *n* = 23. Scale bars represent 50 µm. (**D**) Distribution of low/medium/high GPR18 expression in extravillous trophoblasts and decidual cells across each placental sample.

**Figure 2 ijms-20-04402-f002:**
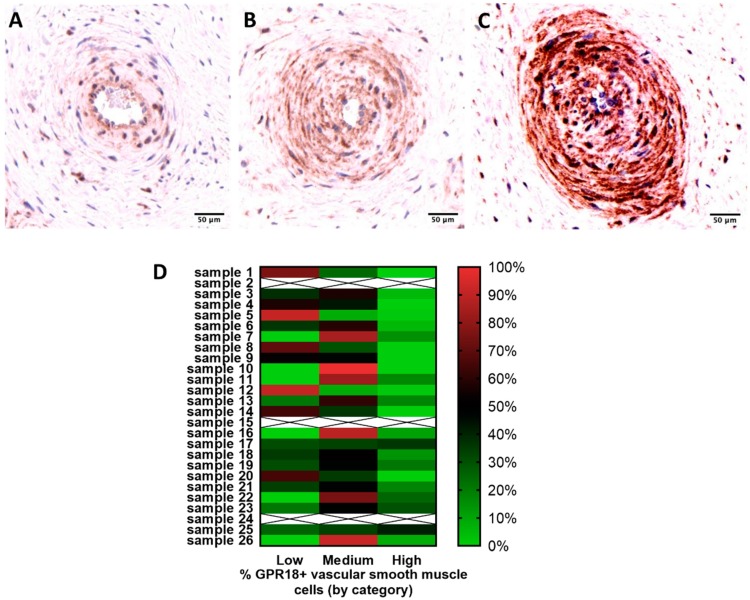
GPR-18 expression in placental vascular smooth muscle. Low (**A**), medium (**B**) and high (**C**) expression of GPR-18 in vascular smooth muscle cells of human term placenta. The images shown are representative, *n* = 23. Scale bars represent 50 µm. (**D**) Distribution of low/medium/high GPR18 expression in vascular smooth muscle cells across each placental sample.

**Figure 3 ijms-20-04402-f003:**
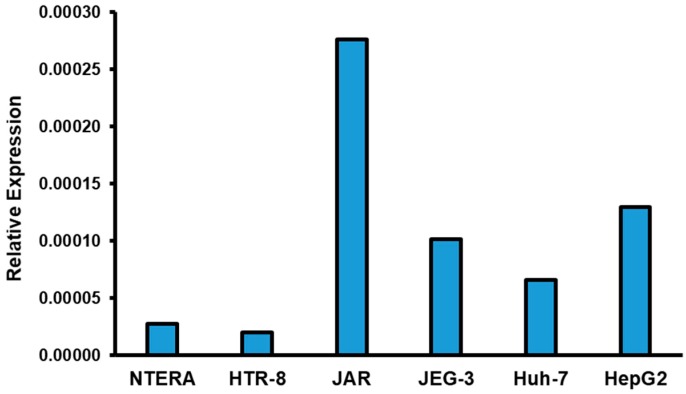
*GPR18* Transcript-level expression in different cell types. Total RNA was isolated and relative expression of *GPR18* was measured in testicular-derived cell line (NTERA), placental trophoblasts (HTR-8, JAR, JEG-3) and hepatocytes (Huh7 & HepG2). Choriocarcinoma-derived placental trophoblasts showed increased expression of *GPR18* compared to other cells types tested (*n* = 1). 18S was used as a control RNA.

**Figure 4 ijms-20-04402-f004:**
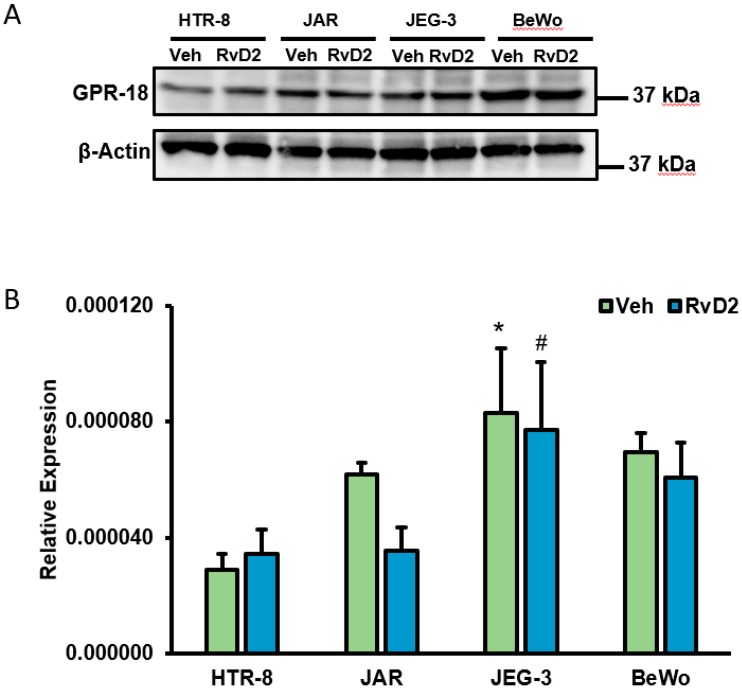
Differential expression analysis of GPR-18 with Resolvin D2 (RvD2) treatment. (**A**) Immunoblot analysis of placental trophoblasts (HTR-8, JAR, JEG-3 and BeWo cells) with 100 nM of RvD2 treatment for 24 h showed slight increase in the protein expression levels of GPR-18 in HTR-8 cells, however GPR-18 protein levels were unaltered with RvD2 treatment in JAR, JEG-3 and BeWo cells compared to actin as control loading. (**B**) Relative expression of GPR18 in placental trophoblasts with 100 nM RvD2 treatment for 24 h. 18S was used as a control RNA. Data are mean ± standard deviation, *n* = 4 experiment. * *p* < 0.05 vs. HTR-8 vehicle treated cells; ^#^
*p* < 0.05 vs. HTR-8 RvD2-treated cells; statistical comparison by ANOVA with post hoc Bonferroni correction.

**Figure 5 ijms-20-04402-f005:**
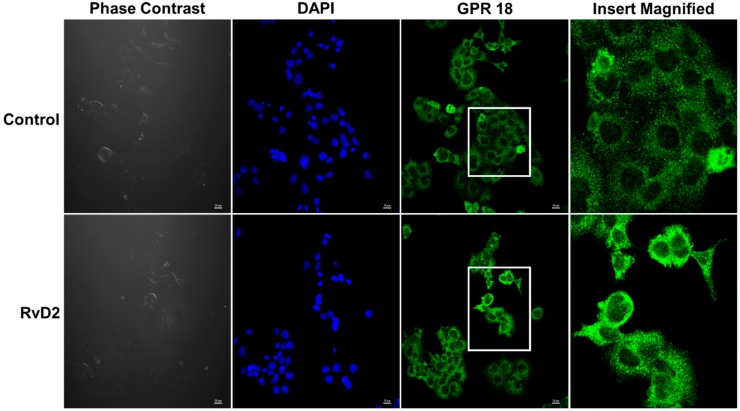
Immunofluorescence analysis of GPR-18 expression in Placental trophoblast with RvD2. Jar cells were treated with 100 nM of resolvin D2 (RvD2) for 24 h and to test their membrane localization. We found an increased plasma membrane expression of GPR-18 in cells treated with RvD2 comparted to control placental trophoblasts. Scale bars represent 20 µm.

**Figure 6 ijms-20-04402-f006:**
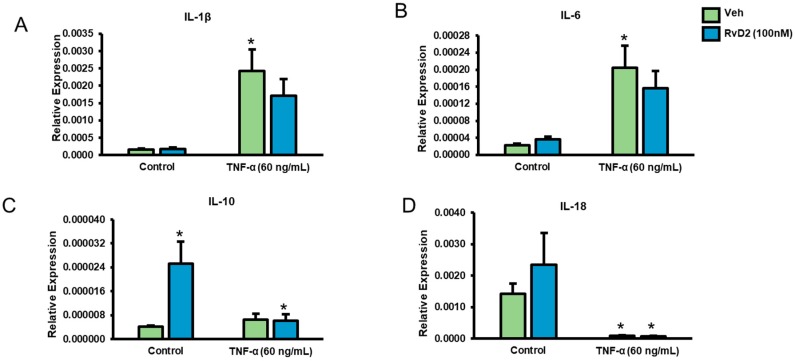
Effect of RvD2 on pro-inflammatory IL-1β and IL-6 transcriptional expression. Total RNA was isolated from placental trophoblasts with and without 100 nM of RvD2 and TNFα (60 ng/mL) treatment and *IL-1β* (**A**), *IL-6* (**B**), *IL-10* (**C**) and *IL-18* (**D**) mRNA expression were measured relative to 18S control RNA. Data are mean ± standard deviation, *n* = 4 experiment. * *p* < 0.05 compared to unstimulated, vehicle-treated (no TNFα/RvD2) cells; statistical comparison by ANOVA with post hoc Bonferroni correction.

**Figure 7 ijms-20-04402-f007:**
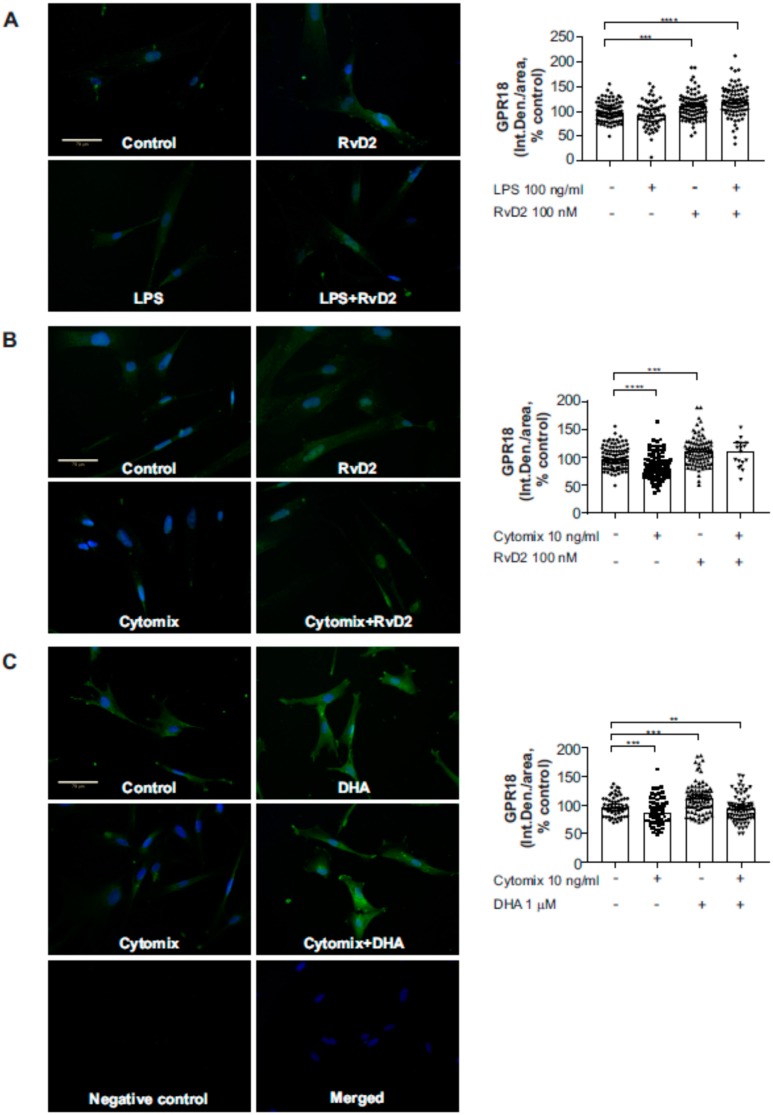
Protein expression of GPR18 upon treatment with resolvin D2. HUASMC were pretreated with RvD2 (**A**,**B**) or DHA (**C**) at indicated concentrations, and then stimulated with LPS (100 ng/mL) (**A**) or cytomix (IL1β, TNFα, IFNγ) (**B**,**C**) for 24 h. At the end of each experiment, cells were fixed and stained for GPR18. Quantification of the immunofluorescent data was performed using ImageJ (**A**–**C**, right panels). Data are median with 95% CI, *n* = 3 experiments except for *n* = 4–5 experiments for control and cytomix treatments on panel C. Data are median with 95% CI, * indicates *p* < 0.05, ** *p* < 0.01, *** *p* < 0.001, **** *p* < 0.0001. Scale bars represent 70 µm.

**Figure 8 ijms-20-04402-f008:**
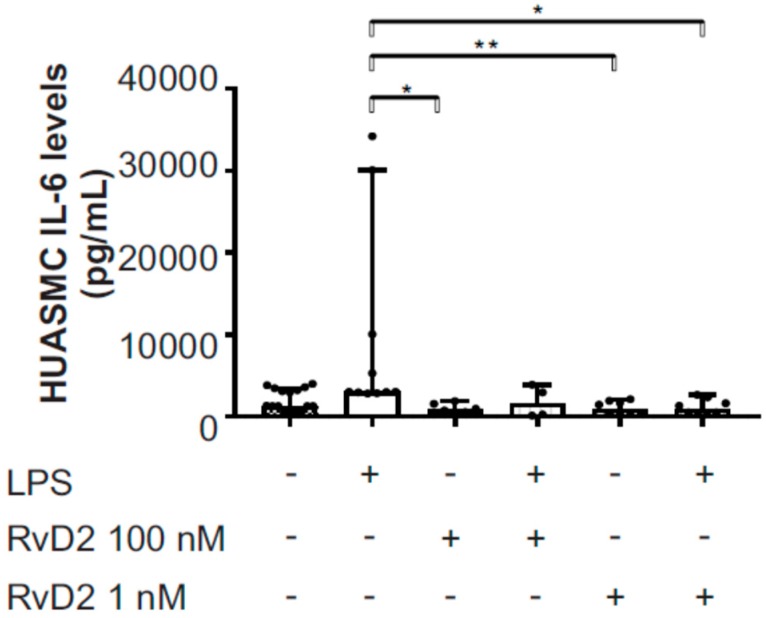
Effects of RvD2 treatment on HUASMC inflammatory cytokine production following LPS treatment. Following one-hour pretreatment with RvD2 at indicated concentrations, HUASMC cells were treated with LPS (100 ng/mL), and IL-6 levels were measured using an ELISA kit after 24 h. Data are from three separate experiments and data are median with 95% CI, * indicates *p* < 0.05, ** *p* < 0.01.

**Figure 9 ijms-20-04402-f009:**
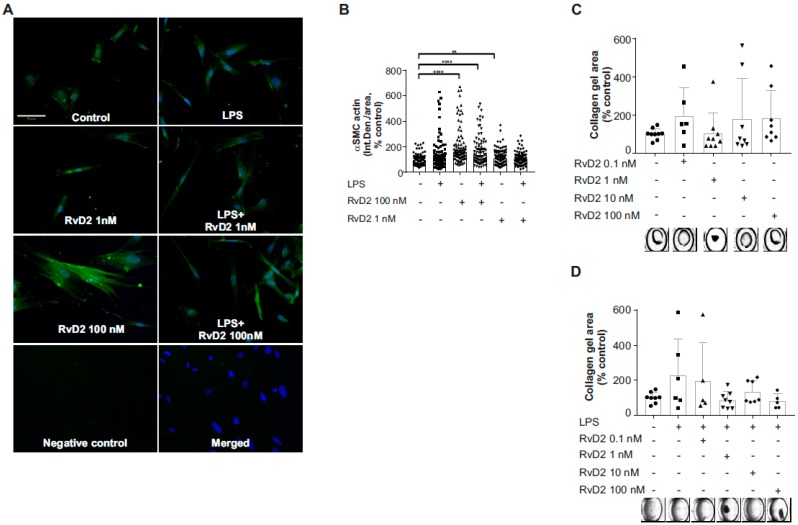
Effects of RvD2 on cytoskeletal changes in HUASMC. (**A**) Immunofluorescent staining for αSMA is shown. HUASMC cells grown on glass coverslips were pretreated for one hour with RvD2 at indicated concentrations, and then stimulated with LPS (100 ng/mL) for 24 h. At the end of each experiment, cells were fixed and stained for α-SMA. Representative images at 20× magnification are shown. Scale bar represents 70 µm. (**B**) Quantification of the immunofluorescent data using ImageJ is shown. Data are from three separate experiments, median with 95% CI. ** indicates *p* < 0.01, **** *p* < 0.0001. (**C**,**D**) Collagen gel contraction assay is shown as % of control. HUASMC cells were grown at 30 × 10^4^ cells/mL concentration in collagen gel mixture, and then pretreated with RvD2 and stimulated with LPS for 24 h. A total of nine images were obtained for each well, and one combined image was obtained using Keyence Image Stitching Module. Representative images for each well are shown. Data are mean ± standard deviation, *n* = 4–5 experiments.

**Table 1 ijms-20-04402-t001:** Participant Characteristics.

Maternal/Infant Characteristics	N	Mean (SD)
Maternal Age, years	26	27.65 (5.16)
Maternal pre-pregnancy BMI, m/kg^2^ (self-report)	17	29.50 (7.81)
Maternal DHA intake, grams	24	0.15 (0.14)
Maternal total omega-3 fatty acid intake, grams	24	2.18 (0.90)
Birth CGA, weeks	26	39.04 (2.46)
Infant Birthweight, grams	26	3357.31 (668.45)
Infant Birth Head Circumference, cm	25	33.86 (2.72)
Infant Birth Length, cm	25	49.12 (3.80)
	**N**	**%**
Race:		
White	14	53.85
African American	7	26.92
Hispanic/Latino	2	7.69
Asian/Pacific Islander	1	3.85
Native American	0	0
Other	2	7.69
Infant Sex (M/F)	17 male 9 female	65.38 34.62
Maternal n-3 Supplement Use (Y/N)	8	30.77
Infant NICU Admission at Birth (Y/N)	8	30.77
Preterm Delivery (<37 weeks gestation) (Y/N)	3	11.54
Suspected or confirmed chorioamnionitis (Y/N)	1	3.85

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
