# Peer review of "Omega-3 Fatty Acid-Derived Resolvin D2 Regulates Human Placental Vascular Smooth Muscle and Extravillous Trophoblast Activities"

_ijms, 2019, doi:10.3390/ijms20184402_

Round 1
Reviewer 1 Report
The manuscript was well written and I have no suggestions but to recommend its acceptance with no changes. The research is novel and original. The authors did a comprehensive review of literature and their research filled some gaps of information in the literature. Therefore, I strongly recommend publication of their results.
Author Response
Reviewer 1:
Comment #1: The manuscript was well written and I have no suggestions but to recommend its acceptance with no changes. The research is novel and original. The authors did a comprehensive review of literature and their research filled some gaps of information in the literature. Therefore, I strongly recommend publication of their results.
Response
We are pleased to see the reviewer’s comments on our manuscript and we believe that our manuscript provides valuable information to the omega-3 research field.
Reviewer 2 Report
I read with great interest the Manuscript titled “Omega-3 Fatty Acid-Derived Resolvin D2 Regulates Human Placental Vascular Smooth Muscle and Extravillous Trophoblast Activities” (ijms-575133). The topic of this manuscript falls within the scope of International Journal of Molecular Sciences. I was particularly pleased to review this paper. In my honest opinion, the topic is interesting enough to attract the readers’ attention. Methodology is accurate and conclusions are supported by the data analysis. Nevertheless, authors should clarify some point, improve the discussion citing relevant and novel key articles about the topic and discuss limitations of the study that are not evidenced in the manuscript.
In general, the Manuscript may benefit from several minor revisions, as suggested below:
How was the number of analyzed placentas defined? Moreover, I would suggest better reporting inclusion and exclusion criteria for included placentas. Were some exclusion criteria adopted? Were some placentas excluded? Was this study registered? I could not find any information about this point. The authors have not adequately highlighted the strengths and limitations of their study. I suggest better specifying these points. Accumulating evidence suggests a clear role for Resolvin RvD2 as a trigger for anti-inflammatory switch (PMID: 30443715). Considering the interesting data on placental trophoblast provided by the authors in this article, do they think that RvD2 may play a role also for classic placental diseases with epigenetic pro-inflammatory fingerprint, such as pre-eclampsia and intrauterine fetal growth restrictions (refer to: PMID: 28243732; PMID: 28282763; PMID: 28466013)?
Author Response
Reviewer 2:
Comment#1:
I read with great interest the Manuscript titled “Omega-3 Fatty Acid-Derived Resolvin D2 Regulates Human Placental Vascular Smooth Muscle and Extravillous Trophoblast Activities” (ijms-575133). The topic of this manuscript falls within the scope of International Journal of Molecular Sciences. I was particularly pleased to review this paper. In my honest opinion, the topic is interesting enough to attract the readers’ attention. Methodology is accurate and conclusions are supported by the data analysis. Nevertheless, authors should clarify some point, improve the discussion citing relevant and novel key articles about the topic and discuss limitations of the study that are not evidenced in the manuscript.
Response:
As suggested by the reviewer, we have improved the manuscript with citations from recent publications on the role of SPMs during pregnancy.
Comment #2:
In general, the Manuscript may benefit from several minor revisions, as suggested below:
How was the number of analyzed placentas defined?
Response:
Placental samples were collected over two years as part of IRB study protocol “Fatty Acids, Fat Soluble Vitamins, Infant Feeding, and Inflammation during NICU Hospitalization” under IRB approval from the University of Nebraska Medical Center (IRB #112-15-EP). This study is intended as a cross sectional study to assess dietary factors impacting neonatal inflammation, and our chosen number of analyzed placenta samples was based on pilot data of staining intensity for GPR18 in HUASMC of the sample tissues. These studies of placental GPR18 expression were not powered to detect significant alterations in staining intensity based on clinical outcomes, although they will inform future studies in this regard. We have now included additional information of our inclusion/exclusion criteria and IRB protocol in the manuscript.
Comment #3:
I would suggest better reporting inclusion and exclusion criteria for included placentas. Were some exclusion criteria adopted?
Response
We thank the reviewer for raising an important point and have now added this information to the manuscript. Exclusion criteria included infants with congenital abnormalities, inborn errors of metabolism, gastrointestinal, liver, or kidney disease, and neonates that were anemic or needed blood transfusions. Infants whose parents were less than 19 years of age were also excluded from the study to avoid obtaining parental consent from a parent who is not legal age in the state of Nebraska. Infants who were made wards of the State of Nebraska were excluded from the study, per Nebraska State law 390 NAC 11-002.04K which states wards of the state may not participate in medical research unless a specific exception is granted by the state, following an evaluation of the protocol by staff in the Medical and Legal Divisions of the Nebraska Health and Human Services System.
Comment #4:
Were some placentas excluded? Was this study registered? I could not find any information about this point.
Response
Following enrollment in the study (using exclusion criteria defined above), collected placentas were excluded if damaged or disrupted upon collection. In addition, we have indicated the reasons of excluding the three samples for GPR18 staining in the original manuscript, and further clarified our description in the methods, section 4.3.
This study was approved by the UNMC IRB Ethics Committee on 04/01/2016 (IRB #112-15 EP, Fatty Acids, Fat Soluble Vitamins, Infant Feeding, and Inflammation during NICU Hospitalization). We have now included this information in our revised manuscript. However, since our study is not a clinical trial, registration at clinicaltrials.gov was not required.
Comment #5:
The authors have not adequately highlighted the strengths and limitations of their study. I suggest better specifying these points. Accumulating evidence suggests a clear role for Resolvin RvD2 as a trigger for anti-inflammatory switch (PMID: 30443715). Considering the interesting data on placental trophoblast provided by the authors in this article, do they think that RvD2 may play a role also for classic placental diseases with epigenetic pro-inflammatory fingerprint, such as pre-eclampsia and intrauterine fetal growth restrictions (refer to: PMID: 28243732; PMID: 28282763; PMID: 28466013)?
Response
We thank the reviewer for the great suggestions to improve our manuscript. We have modified our discussion in the revised manuscript and incorporated the publications as suggested by the reviewer in the discussion.
We added the following statement on the limitations of our study in Discussion (highlighted in red), as shown below:
‘While our findings provide insight into the role of SPMs in placental tissue, our study is not without limitations. This is intended as a cross-sectional study mainly focusing on the measurement of Resolvin D2 and its receptor in trophoblasts and vascular smooth muscle cells of the placenta; however, the fluctuations in SPMs during different trimesters of pregnancy and how this might contribute to fetal health is unknown. Also, while we show in placental cells that RvD2 is involved in reducing inflammation, we are unable to explain any causal relationships regarding RvD2 and preterm births.’
Reviewer 3 Report
Page 4, line “Vascular smooth muscle cell GPR18 immunoreactivity was also measured in 23 placental tissue”. Based on the table 1 there’s 26 sample in total. Please state why there are 3 samples being exclude from this analysis. The sample size for cell experiments are too small (many data were from only 2 experiments). With the small sample size that have been used in many of the experiments in this paper, the statistical method was not appropriate. When the sample size is smaller than n=8-10, non-parametric analysis should be utilized in the analysis and data should be expressed as median ± Standard deviation. The data analysis and presentation are not appropriate in figure 8 and figure 9. The data variation is very high, please make sure the correct statistically analysis method was used. As the author mentioned that the experiment was repeat 3 times and each treatment are with more than 3 data point. If there are multiple measurement taken in each experiment the mean/or median value of each experiment should be presented. Figure 7 & figure 9, there are no negative control for the staining. Figure 7, the n=2 experiment is very small sample size. Data from only 2 experiments in the cells it is not conclusive. Please increase the n number (a minimum of n=3~4 is required). A large portion of the 2nd paragraph in the discussion is actually back ground information which should be include in the introduction instead. Please focus the discussion on the findings in the current study.
Author Response
Reviewer 3:
Comment #1:
Page 4, line “Vascular smooth muscle cell GPR18 immunoreactivity was also measured in 23 placental tissue”. Based on the table 1 there’s 26 sample in total. Please state why there are 3 samples being exclude from this analysis.
Response
We have added additional clarification as to the exclusion of the three samples for GPR18 staining that we had included in the original manuscript, including in Section 4.3 as below:
‘As stated in Methods, three cases were excluded from the analyses (2 had less than 10 blood vessels and one exhibited poor staining).’
Comment #2:
The sample size for cell experiments are too small (many data were from only 2 experiments).
Response
As suggested by the reviewer, we performed additional experiment to reach n=3 experiments in cell experiments and included in the revised manuscript.
Comment #3:
With the small sample size that have been used in many of the experiments in this paper, the statistical method was not appropriate. When the sample size is smaller than n=8-10, non-parametric analysis should be utilized in the analysis and data should be expressed as median ± Standard deviation. The data analysis and presentation are not appropriate in figure 8 and figure 9. The data variation is very high, please make sure the correct statistically analysis method was used. As the author mentioned that the experiment was repeat 3 times and each treatment are with more than 3 data point. If there are multiple measurement taken in each experiment the mean/or median value of each experiment should be presented.
Response
We followed reviewer’s suggestion on statistical analysis. We now use non-parametric data analysis (Wilcoxon -Mann- Whitney Test). Since we are also interested in cell to cell variation in either GPR18 or aSMC expression, we prefer to keep the data as plotted. Fluorescence cell images are routinely analyzed using fluorescent intensity/ area, and in general each cell is considered when doing statistics (Carr et al, Mol. Cell. Biol. 2013, 33(3):622.; Ulu et al, J Cell Sci , J Cell Sci. 2018 Feb 1;131(3). and Song et al J Cell Sci, J. Cell. Sci., 128(5):913-22). We now clarified our data analysis in Methods. Briefly, images were obtained from at least five different fields for each treatment in each of the three independent experiments. Statistical analysis of these experiments was performed by comparing all values from 3 experiments among treatment groups. These data were tested for normality using Shapiro-Wilk test, and those that did not pass normality was analyzed using Wilcoxon-Mann-Whitney non-parametric test.
Further, in figure 8, we kept the IL-6 data presentation as it is, because data were obtained by running two biological replicates for each of the separate ELISA experiments. However, as suggested by the reviewer we analyzed our data for normality and performed non-parametric Kruskal-Wallis test.
Comment #4:
Figure 7 & figure 9, there are no negative control for the staining.
Response
We have now included negative controls (no primary antibody control) for each staining as suggested.
Comment #5:
Figure 7, the n=2 experiment is very small sample size. Data from only 2 experiments in the cells it is not conclusive. Please increase the n number (a minimum of n=3~4 is required).
Response
We ran additional experiments and now included at least n=3-4 in Figure 7A and B as well as for Figure 9.
Comment #6:
A large portion of the 2nd paragraph in the discussion is actually back ground information which should be include in the introduction instead. Please focus the discussion on the findings in the current study.
Response
As suggested, we moved that background information to the introduction.
Round 2
Reviewer 3 Report
The authors addressed most of the points being brought up by the reviewer.